# Ag–thiolate interactions to enable an ultrasensitive and stretchable MXene strain sensor with high temporospatial resolution

Yang Liu[1,5] ✉, Zijun Xu[1,5], Xinyi Ji [2,5], Xin Xu[1], Fei Chen[1], Xiaosen Pan [1], Zhiqiang Fu[1], Yunzhi Chen[1], Zhengjian Zhang[1], Hongbin Liu[1], Bowen Cheng[1] ✉ & Jiajie Liang [2,3,4] ✉

High-sensitivity strain sensing elements with a wide strain range, fast response, high stability, and small sensing areas are desirable for constructing strain sensor arrays with high temporospatial resolution. However, current strain sensors rely on crack-based conductive materials having an inherent tradeoff between their sensing area and performance. Here, we present a molecular-level crack modulation strategy in which we use layer-by-layer assembly to introduce strong, dynamic, and reversible coordination bonds in an MXene and silver nanowire-matrixed conductive film. We use this approach to fabricate a crack-based stretchable strain sensor with a very small sensing area (0.25 mm$^2$). It also exhibits an ultrawide working strain range (0.001–37%), high sensitivity (gauge factor ~500 at 0.001% and >150,000 at 35%), fast response time, low hysteresis, and excellent long-term stability. Based on this high-performance sensing element and facile assembly process, a stretchable strain sensor array with a device density of 100 sensors per cm$^2$ is realized. We demonstrate the practical use of the high-density strain sensor array as a multichannel pulse sensing system for monitoring pulses in terms of their spatiotemporal resolution.

Strain sensors based on crack generation and propagation mechanisms hold great promise for flexible mechanosensing designs and provide ultrahigh sensitivity and capabilities in a variety of performance-demanding and intelligent applications[1–6]. Some sophisticated applications, such as the fingertips of humanoid robots and implantable or skin-mounted health monitoring electronics, require stretchable mechanosensing arrays that can sense and identify complex signals with both high sensing accuracy and temporospatial resolution[7–10]. However, the development of such stretchable strain sensor arrays with the required sensitivity, strain range, stability, and spatiotemporal resolution remains a formidable challenge. In recent years, enormous research efforts have been made to overcome the tradeoff between sensitivity and strain range in a single strain sensor[6,11–13]. However, little progress has been made to achieve integrated sensor arrays or systems with a high spatiotemporal resolution. Most strain sensor arrays demonstrated in the literature are based on the integration of bulky sensing elements with footprints > 10 mm$^2$ [14–16]. These large dimensions and low sensor densities are incompatible with highly compact integration and prevent the realization of sensor arrays with high spatiotemporal resolution.

To achieve high-density sensor arrays, it is necessary to shrink the footprint of single sensor elements down to the millimeter-scale or

[1]State Key Laboratory of Biobased Fiber Manufacturing Technology, Tianjin Key Laboratory of Pulp and Paper, Tianjin University of Science and Technology, Tianjin, China. [2]School of Materials Science and Engineering, National Institute for Advanced Materials, Nankai University, Tianjin, China. [3]Key Laboratory of Functional Polymer Materials of Ministry of Education, College of Chemistry, Nankai University, Tianjin, China. [4]School of Materials Science and Engineering & Smart Sensing Interdisciplinary Science Center, Nankai University, Tianjin, China. [5]These authors contributed equally: Yang Liu, Zijun Xu, Xinyi Ji. ✉e-mail: liuyangtust@tust.edu.cn; bowen15@tiangong.edu.cn; liang0909@nankai.edu.cn

even microscale. However, the miniaturization of the active size in crack-based strain sensors is difficult to achieve due to the inherent tradeoff between sensing area and performance. When a brittle conductive sensing film is stimulated by a small strain, cracks could be generated, which will propagate perpendicularly to the strain direction. This may significantly disrupt conductive pathways, resulting in a high sensitivity at small strains[1,17,18]. However, cracks are difficult to control and easily expand, leading to cut-through patterns and subsequent mechanical fracture in sensing films[19,20]. As such, the crack-based strain sensors usually suffer from a narrow strain range and poor reliability[19–21]. According to the crack-based thin-film principle[6,22], larger films can generate more cracks and prolong the crack propagation to accommodate a greater applied strain[6]. Thus, the footprints of most reported crack-based strain sensors have been on the order of millimeters or even centimeters, which inevitably leads to both a low sensor density and sensing resolution in integrated sensor arrays[23]. Another commonly used strategy to optimize the strain-sensing performance of crack-based sensors involves regulating the cracking process in the sensing film. Several crack engineering methods, such as bridging cracks, stepwise cracks, interlayer insertion, bio-mimetic architecting, film wrinkling, and substrate structuring[3,5,22,24], have been proposed to improve the strain range and sensing stability of crack-based strain sensors. However, these advancements usually come at the cost of sensitivity and detectability over a small strain range[6,25]. This is because these crack modulation strategies usually generate small or short cracks and may even cause buckling instead of long cracks in conductive sensing films under tiny strain, which may save conductive pathways[22]. Thus, current crack-based strain sensors still face formidable difficulties in simultaneously achieving a high sensitivity, wide strain range, high stability, fast response time, small size, and high spatiotemporal resolution[26].

According to crack-based sensing mechanisms[3,5,22], to achieve these strain-sensing properties simultaneously, the structure and cracking mode in the sensing film should include several features. (1) To guarantee a high sensitivity, the sensing film should generate long (or cut-through) cracks to significantly disrupt the conductive pathways. (2) To minimize the sensing area, the sensing film should produce dense mesh-like cracks that reduce the size of fragmented conductive islands. (3) To improve the strain range, the sensing film should exhibit an effective behavior to energy dissipation to accommodate large applied strain. (4) To obtain high reliability, the deformation of the sensing structure should be reversible during repeated stretch-release cycles.

Here, we propose a cracking control strategy by introducing strong, dynamic, and reversible coordination bonds in an inorganic-matrixed conductive film. Using this approach, we prepare a cracked-based stretchable strain sensor with minimal sensing size and good all-round sensing performance. A sensing film was fabricated from thiolate-terminated MXene (S-MXene) nanosheets and silver nanowire (AgNW) via a layer-by-layer technique to form robust and dynamic thiol–silver (S–Ag) coordination bonds that were distributed in the multilayered sensing thin-film. When the multilayered sensing film was stretched, long cracks were first produced and propagated through weak interfaces or boundaries in the inorganic-matrixed film. The strong S–Ag bonds regulated local strain fields in the brittle matrix, which forced cracks to deflect and twist, resulting in dense mesh-like cracking in the sensing film. The dynamic S–Ag bonds enabled the multilayer matrix to exhibit energy dissipation that prohibited crack gaps from opening. This maintained nanoscale crack gaps within the film, even under a large strain. The reversible S–Ag bonds allowed the crack patterns to heal after unloading, which improved the structural stability of the sensing film. Using this cracking control design, the S-MXene/AgNW strain sensor (S-M/A) with a minimal sensing area of 0.25 mm² simultaneously provided an ultrawide strain sensing range (0.001–37%), ultrahigh sensitivity (gauge factor (GF) > 500 at 0.001% and > 150,000 at 35%), rapid response (~5 ms), low hysteresis, and high stability. The sensing area was more than one order of magnitude smaller than that of all previously reported crack-based strain sensors. This small but effective sensing area enabled us to fabricate a stretchable strain sensor array with a device density of 100 sensors per square centimeter, thus achieving a high spatiotemporal sensing resolution. This work realizes all of the desirable sensing properties (i.e., a high resolution, range, sensitivity, speed, and stability) in a single strain sensor system, which are particularly attractive for wearable and intelligent applications that require monitoring highly precise and feature-rich signals.

## Results

We obtained the S-MXene nanosheets via a hydrothermal reaction between 3-mercaptopropyltriethoxysilane (MPTES) and MXene (described in detail in the experimental section). Then, we fabricated the S-M/A strain sensing film through the sequential adsorption of S-MXene nanosheets and AgNW on a glass substrate using an assembly process in solution, followed by transferring into a stretchable polyurethane substrate (Fig. 1a). The thiol groups on the S-MXene reacted with AgNW to form S–Ag bonds distributed at the interfaces between layers of the S-M/A film[27]. Because the S–Ag coordination bond energy (about 17.7 kcal/mol)[28] is much higher than that of the hydrogen bonds (F···H) between MXene nanosheets (2–3.2 kcal/mol)[25], they can serve as strong crosslinking points in the

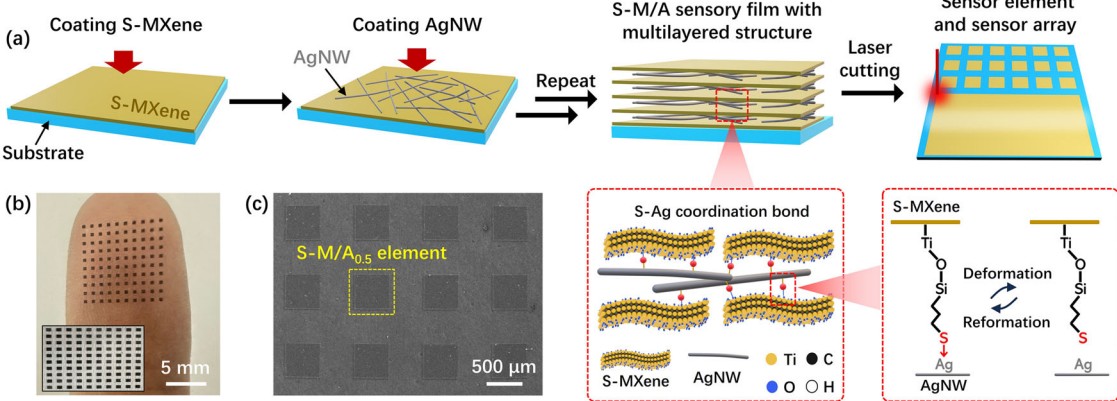

**Fig. 1 | Introduction of S–Ag coordination bonds into S-M/A sensing films. a** Schematic illustration of the fabrication process of an S-M/A sensing film and sensor array. Inset of the mechanism for dynamic and reversible S–Ag coordination bonds formed between S-MXene and AgNW. **b** Photograph of a strain sensor array with 10 × 10 S-M/A$_{0.5}$ sensing elements on a fingertip (25-year-old male volunteer). The inset shows the sensor array under 30% strain. **c** Top-view SEM image of the S-M/A$_{0.5}$ sensing elements.

S-M/A film. The coordination mode of S−Ag bonds can be reversibly switched on/off to permit energy dissipation and self-healing in the S-M/A film[29,30]. A sensor element with a small active area and a corresponding sensor array with a high sensor density were obtained through a one-step laser etching process applied on the as-prepared S-M/A film. S-M/A sensors with different sensing dimensions, $0.5 \times 0.5\ mm^2$, $1 \times 1\ mm^2$, $2 \times 2\ mm^2$, and $5 \times 5\ mm^2$ were fabricated and named $S\text{-}M/A_{0.5}$, $S\text{-}M/A_1$, $S\text{-}M/A_2$, and $S\text{-}M/A_5$, respectively (Fig. 1b and Supplementary Fig. 1). As can be observed from the scanning electron microscopy (SEM) image (Fig. 1c), the laser-etched sensing units had clear and smooth edges, indicating the complete etching and high efficiency of the laser etching method. Using this process, we fabricated a strain sensing array composed of 100 $S\text{-}M/A_{0.5}$ units, showing a device density of 100 sensors per $cm^2$ (Fig. 1b, c). For comparison, crack-based strain sensors based on pure MXene nanosheets and AgNW were fabricated (named M/A) with dimensions of $0.5 \times 0.5\ mm^2$, $1 \times 1\ mm^2$, $2 \times 2\ mm^2$, and $5 \times 5\ mm^2$ (named $M/A_{0.5}$, $M/A_1$, $M/A_2$, and $M/A_5$, respectively).

The prepared S-MXene nanosheets were first characterized by energy-dispersive X-ray spectroscopy (EDS), X-ray diffraction (XRD), Fourier-transform infrared (FT-IR) spectroscopy, X-ray photoelectron spectroscopy (XPS), and water contact angle measurements. EDS confirmed the existence and uniform distribution of S atoms over the S-MXene surface (Fig. 2a). As shown in Fig. 2b, two new peaks at $2580\ cm^{-1}$ and $944\ cm^{-1}$, ascribed to −SH stretching vibrations and Ti−O−Si asymmetric stretching, appeared in the FT-IR spectrum of S-MXene[31,32]. No peak in Ti−O−Si asymmetric stretching could be seen in MPTES and pure MXene. The XPS of S-MXene revealed a typical Ti−O−Si bond at 531.7 eV in O $1s$ spectra (Supplementary Fig. 2a) and Si−O−C at 102.2 eV in Si $2p$ spectra (Supplementary Fig. 2b)[33], proving the successful grafted of MPTES onto MXene. The XRD patterns of the S-MXene and MXene films exhibited peaks at 6.6° and 7.0°, respectively, confirming the modification of thiol groups onto the MXene nanosheets (Fig. 2c). Moreover, the water contact angle increased from 60.2° for MXene to 88.5° for S-MXene (Fig. 2d), indicating the lower water-wettability of the S-MXene film. Combined, these results indicate the successful synthesis of S-MXene nanosheets. The thiolate surface group in the S-MXene could block the contact between MXene and water or dissolved oxygen and prohibit the oxidation reaction[34-39], thus significantly improving the stability of S-MXene in water and oxygen environment (Supplementary Fig. 3).

As shown in the cross-sectional SEM image (Fig. 2e), the obtained S-M/A sensing film exhibited a multilayered structure with alternating layers of S-MXene nanosheets and AgNW networks. The thickness of the S-M/A sensing film was controlled by the number of coatings (Supplementary Fig. 4). The optimal film thickness was optimized and fixed at about $1.4\ \mu m$ (Supplementary Fig. 5 and Supplementary Note 1). The top-view SEM and EDS characterizations confirmed the uniform distribution of AgNW on the S-MXene surface (Supplementary Fig. 7). XPS characterization validated the formation of Ag-S coordination bonds between AgNW and S-MXene. Ag $2d_{5/2}$ (368.4 eV) and Ag $2d_{3/2}$ (374.4 eV) peaks in AgNW films shifted to 367.1 eV and 373.1 eV in the XPS spectrum of the S-M/A film (Fig. 2f), respectively, due to Ag−S coordination bonds between AgNW and S-MXene[40,41]. As shown in the UV-vis spectrum (Fig. 2g), the Ag characteristic peaks red-shifted from 350 nm and 380 nm to 352 nm and 384 nm, which was attributed to Ag−S coordination bonds, in line with previously reported results[41,42]. In addition, a broad Ag−S stretching mode appeared near $220\ cm^{-1}$ in the Raman spectrum of the S-M/A film (Fig. 2h)[40]. All of these results suggest the formation of Ag−S coordination bonds in the S-M/A sensing films.

To investigate the sensing performance at small strain (<0.1%), the crack-based sensors were periodically strained using bending and then allowed to recover, as shown in Fig. 3a. The strain applied to the sensing devices was precisely calculated from the bending degree of the substrate (Fig. S3). Figure 3b shows cyclic variations in the relative resistance for different tiny strains applied to the $S\text{-}M/A_{0.5}$ sensor. The detectable minimum strain of $S\text{-}M/A_{0.5}$, $S\text{-}M/A_2$, and $S\text{-}M/A_5$ sensors all reached 0.001%, which exceeds the value of most previously reported crack-based strain sensors (Supplementary Table 1)[5,43-50]. The sensitivity (GF) in the small-strain range of 0−0.05% for $S\text{-}M/A_{0.5}$ reached about 500 with linearity of 0.99 (Fig. 3c), which is similar to that of $S\text{-}M/A_1$, $S\text{-}M/A_2$, and $S\text{-}M/A_5$ sensors with larger sensing areas (Supplementary Fig. 9). Figure 3d reveals that the response and recovery times of $S\text{-}M/A_{0.5}$ to an applied strain of 0.005% were 5 ms, which is superior to that of most reported crack-based strain sensors (Supplementary Table 1)[43,46-48,51]. To confirm repeatability, five samples for each device were tested and exhibited similar performance (Supplementary Fig. 10 and 11).

To explore the sensing performance under large strain (>0.1%), the strain sensors were mounted on a motorized linear stage and periodically strained in the stretch-and-release mode (Fig. 3a).

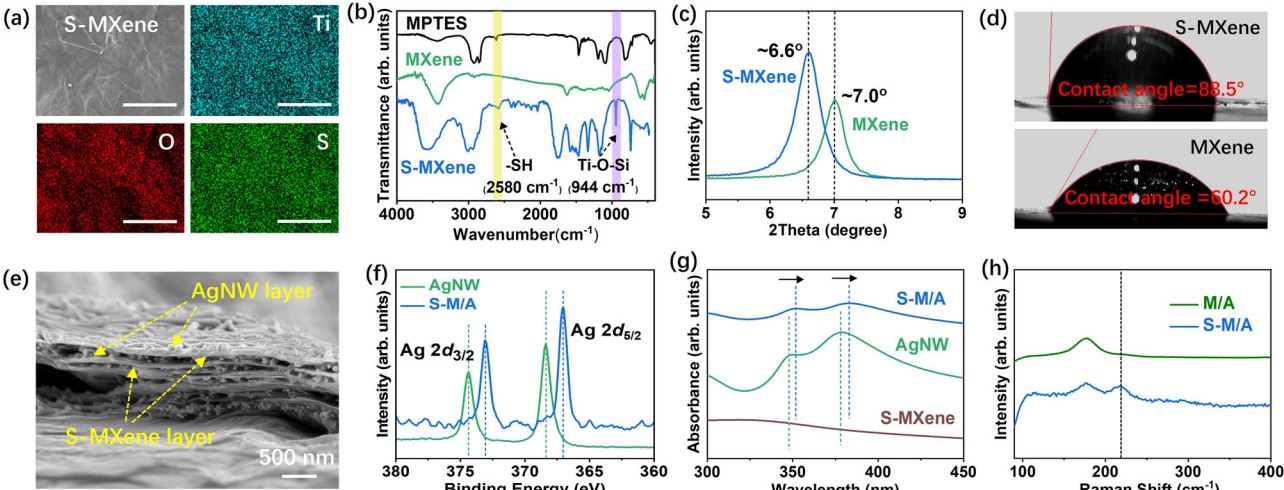

**Fig. 2 | Characterization of S-MXene and S-M/A films. a** SEM image and EDS element maps of Ti, O, and S of S-MXene. Scale bar = 10 μm. **b** FT-IR spectra of MPTES, MXene, and S-MXene. **c** XRD patterns of MXene and S-MXene. **d** Water contact angles of MXene and S-MXene. **e** Cross-sectional SEM image showing the layer-by-layer structure of the S-M/A film. **f** XPS spectra of AgNW and S-M/A films in the Ag $2d$ region. **g** UV-vis spectra of AgNW, S-MXene, and S-M/A films. **h** Raman spectra of M/A and S-M/A films.

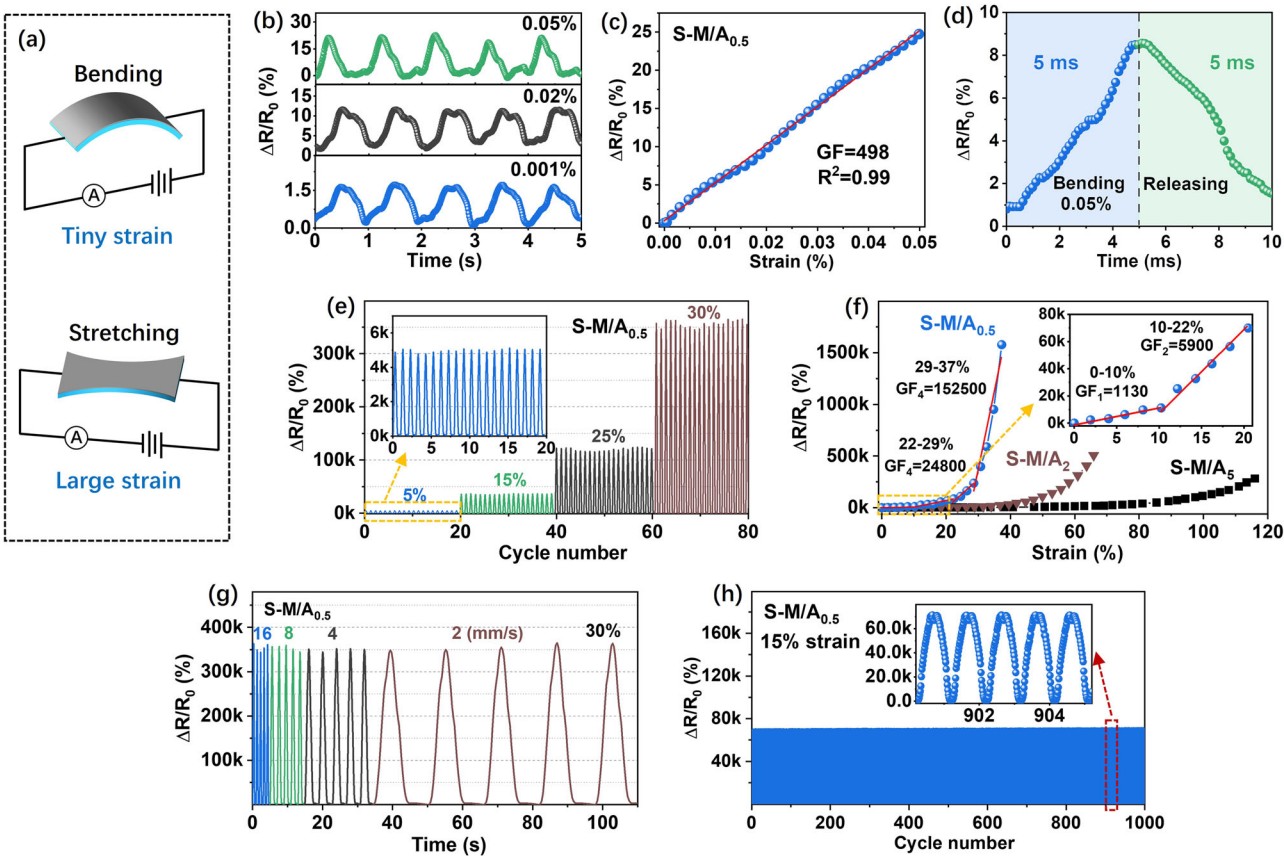

**Fig. 3 | Strain-sensing performance of S-M/A sensors. a** Schematic illustration of the device being bent and stretched to tiny and large strains during sensing performance measurements. **b** Relative resistance changes during the detection of ultralow strains measured using the S-M/A$_{0.5}$ sensor. **c** Relative resistance variation as a function of tiny strain (0–0.05%) for the S-M/A$_{0.5}$ sensor. **d** The transient sensing response and recovery time to an applied strain of 0.05%. **e** Relative resistance changes under various cyclical large strains (5%, 15%, 25%, and 30%) for S-M/A$_{0.5}$ the sensor. **f** Relative resistance variation as a function of large strain for S-M/A$_{0.5}$, S-M/A$_2$, and S-M/A$_5$ sensors. **g** Relative resistance variation of S-M/A$_{0.5}$ sensor under incremental strain rate cycling between 0% and 30% strain. **h** Relative resistance changes of S-M/A$_{0.5}$ sensor to 15% strain over 1000 stretch-release cycles. RH = 50%.

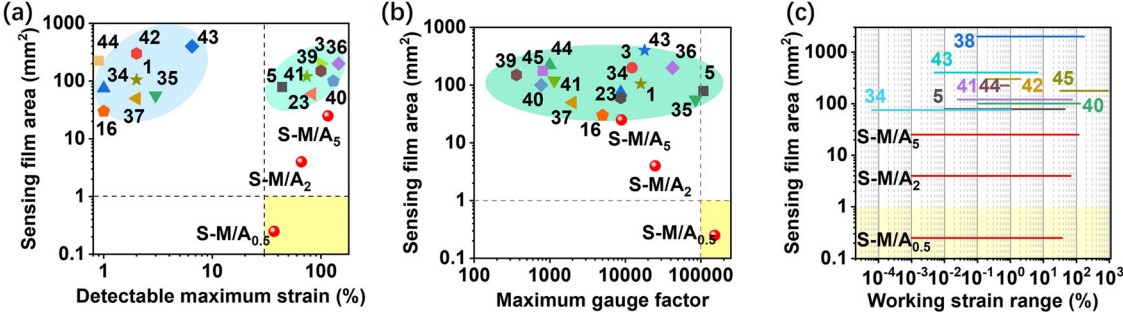

**Fig. 4 | Strain-sensing performance comparison. a** Sensing film dimensions and detectable maximum strain of S-M/A$_{0.5}$, S-M/A$_2$, and S-M/A$_5$ sensors compared with previously-reported crack-based strain sensors[1,3,5,18,25,43–49,51–53,55,56]. **b** Sensing film dimensions and maximum GF of S-M/A$_{0.5}$, S-M/A$_2$, and S-M/A$_5$ sensors compared with previously-reported crack-based strain sensors[1,3,5,18,25,43–49,51–53,55,56]. **c** Sensing film dimensions and working strain range of S-M/A$_{0.5}$, S-M/A$_2$, and S-M/A$_5$ sensors compared with previously-reported crack-based strain sensors[5,43–49,51].

Figure 3e illustrates the relative resistance variation of the S-M/A$_{0.5}$ sensor under various cyclic strains up to 5%, 15%, 25%, and 30% at a strain rate of 2 mm/s, which delivered repeatable and stable relative resistance changes of about 5000; 36,000; 125,000; and 360,000, respectively. As shown in Fig. 3f and Supplementary Fig. 12, the sensing curves of the S-M/A sensors were all subdivided into four linear regions. The S-M/A$_{0.5}$ sensor, with the smallest sensing area of 0.25 mm², exhibited a large strain range of up to 37% with an ultrahigh GF over the entire working strain range: 1130; 5900; 24,800; and 152,500

in the strain ranges of 0–10%, 10–22%, 22–29%, and 29–37%, respectively (Fig. 3f). The GF of 152,500 is one of the highest values ever reported for crack-based strain sensors with a working strain range >30% (Supplementary Table 1)[3,5,25,43–45,49,52,53]. As expected, the working strain range increased and the sensitivity decreased as the sensing area increased for the S-M/A sensors (Supplementary Fig. 12). The S-M/A$_5$ sensor, with a sensing area of 5 × 5 mm², responded to strains as large as 116%, and the GF of the sensor reached 116, 770, 2800, and 8890 in the strain ranges of 0–47%, 47–75%, 75–94%, and 94–116%,

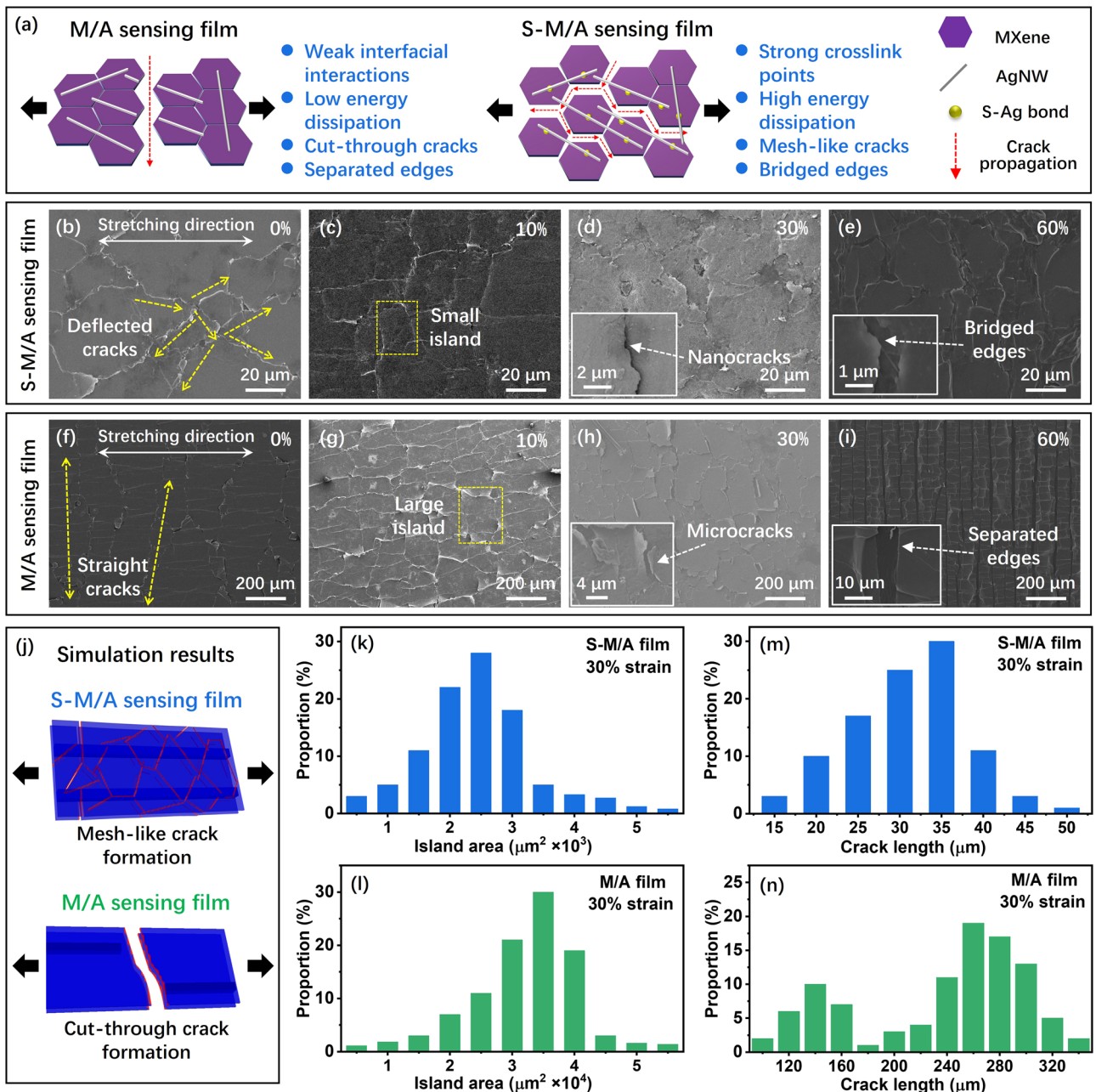

**Fig. 5 | Crack modulation behavior of S-M/A sensing films. a** Illustration of crack formation and propagation in the M/A and S-M/A sensing films. SEM images of crack propagation in **b**–**e** S-M/A film and **f**–**i** M/A film under strain states of 0%, 10%, 30%, and 60%. Inset in **d** the local magnification of nanocracks; inset in **h** the local magnification of microcracks; inset in **e** local magnification of bridged edges; inset in **i** the local magnification of separated edges. **j** ABAQUS finite element simulation of the cracking behavior in M/A and S-M/A films. Histograms of the conductive island area in **k** S-M/A and **l** M/A films under 30% strain. Data were collected from five images under each condition. Histograms of the crack length in **m** S-M/A and **n** M/A films under 30% strain. Data were collected from five images under each condition.

respectively. In contrast, the M/A$_{0.5}$ and M/A$_1$ sensors failed to reliably respond to any strain, while the M/A$_2$ and M/A$_5$ sensors showed much smaller strain ranges than those of the S-M/A sensors (Supplementary Fig. 13).

The sensing stability and reliability of the S-M/A sensors were further examined. All S-M/A$_{0.5}$, S-M/A$_1$, S-M/A$_2$, and S-M/A$_5$ sensors exhibited a relatively small hysteresis during cyclic stretch-release with a high strain (Supplementary Fig. 14), which allowed for durable and reliable dynamic operations[54]. Figure 3g illustrates the output signals of the S-M/A$_{0.5}$ sensor under various strain rates. Under a maximum strain of 30%, the electrical responses of the strain sensors were steady and remained stable, even while increasing the strain rates from 2 mm/

s to 16 mm/s. The S-M/A$_{0.5}$ sensor was subjected to cyclic bending experiments between 0% and 0.05% strain (Supplementary Fig. 15) and cyclic stretching measurements between 0% and 15% strain (Fig. 3h), respectively. During the long-term tests over thousands of strain cycles, both the peak and baseline resistance changes of the S-M/A$_{0.5}$ sensor remained stable. Moreover, the S-M/A$_5$ sensor endured more than 7000 stretch-release cycles with a large peak strain of 70% under different relative humidity (RH) conditions (Supplementary Fig. 16). The S-M/A sensors remained high sensing stability even under a high RH of 85% (Supplementary Fig. 16b). All of these results confirm the good long-term stability, durability, and reliability of our S-M/A strain sensors.

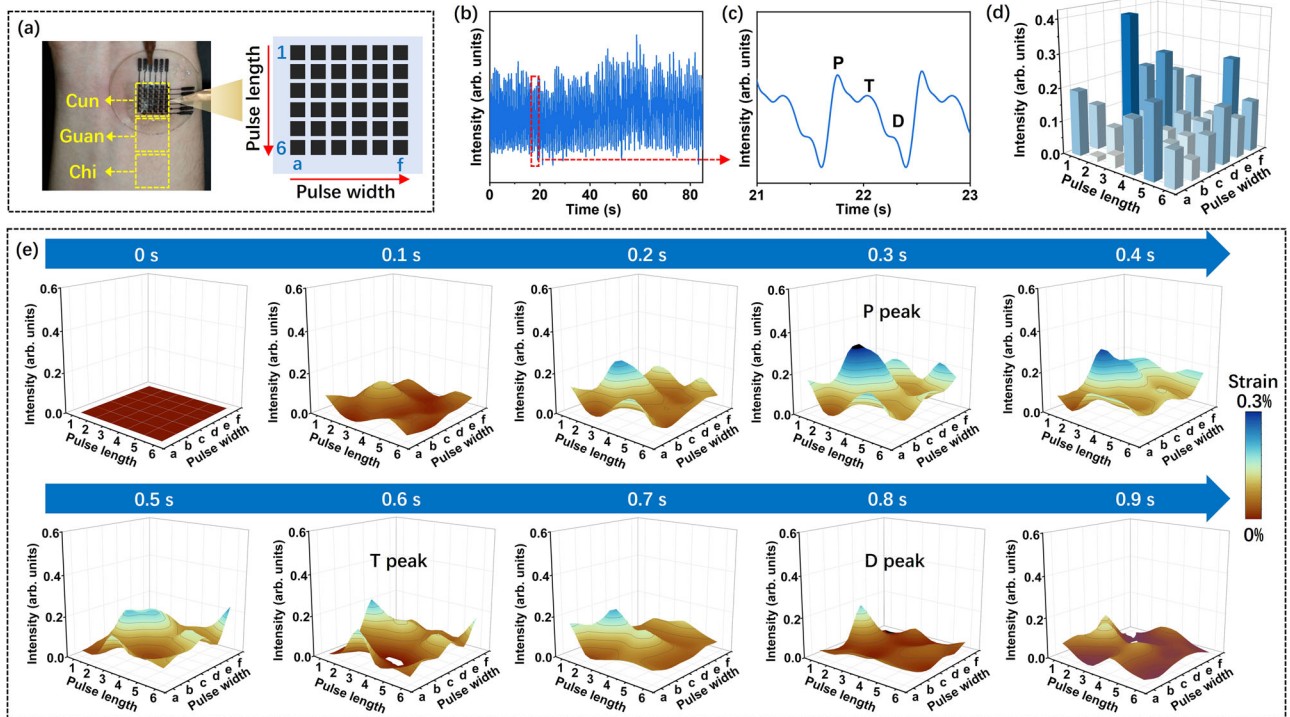

**Fig. 6 | Application of S-M/A sensor array for pulse signal measurements. a** A 36-channel S-M/A$_1$ strain sensor array was placed at the wrist above the radial artery to detect the pulse signals at the position of Cun. Right image: Schematic of the strain sensor array and the position of the 36 channels. Sensing unit positions from 1 to 6 along the pulse length and sensing unit positions from (a) to (f) along the pulse width. **b** Pulse waves are measured continuously by the 3c channel from the sensor array. **c** Amplification of the single pulse signal in (**b**). **d** Column diagrams for pulse strength distribution measured by the 36-channel sensor array. **e** A series of 3D surface diagrams at every 0.1 s in one complete pulse cycle measured by the 36-channel sensor array. The characteristic peaks of *P*, *T*, and *D* appeared at 0.3 s, 0.6 s, and 0.8 s, respectively. The 3D surface was constructed from 36-channel pulse signals by polynomial fitting.

Compared with other state-of-the-art crack-based strain sensors with high GFs (>300) and large working strain ranges (>30%)[3,5,25,43–45,49,52,53], our S-M/A sensors demonstrated several advantages. First, the S-M/A$_{0.5}$ sensor exhibited reliable responses to strains up to 37% with an ultra-small sensing area of 0.25 mm$^2$ (Fig. 4a), which is more than one order of magnitude smaller than all previously reported crack-based strain sensors (Supplementary Table 1)[1,3,5,18,25,43–49,51–53,55,56]. Secondly, our S-M/A$_{0.5}$ sensor achieved ultrahigh GF values (≥500) across the entire working strain range, with a maximum GF of over 152,500 in the 29–37% strain range, which outperforms all reported crack-based sensors (Fig. 4b and Supplementary Table 1). Third, different from the conventional approach of fabricating crack-based strain sensors with sensing dimensions >50 mm$^2$, the working strain range of the S-M/A sensor could be controlled from 0.001% to >110% by changing the sensing area of its sensor elements from 0.25 mm$^2$ to 25 mm$^2$ (Fig. 4c and Supplementary Table 1)[5,43–51].

The sensing properties of the S-M/A sensing films were largely attributed to the presence of strong, dynamic, and reversible S–Ag bonds within their multilayered structures. A model of the cracking behavior is proposed in Fig. 5a to illustrate the strain-regulating effect of the MXene and AgNW-matrixed brittle films. To understand the high sensitivity and wide strain range of the S-M/A$_{0.5}$ sensor with such a small sensing area, morphological changes within the M/A and S-M/A films under various strains were compared using SEM (Fig. 5b–i and Supplementary Figs. 17 and 18). The cracks propagated in a straight line, transverse to the applied strain through weak interfaces inside the M/A film (Fig. 5f). This is consistent with the typical cracking behavior in brittle sensing films[3,22,56]. This inevitably led to the generation of long cut-through cracks and large conductive islands (defined as the area surrounding the cracks), which prevented the dimensions of the sensing film from decreasing and its strain range from increasing[12,24]. In contrast, the strong S–Ag coordination bonds can induce stress concentration in the S-M/A sensing film under strain[23]. When cracks propagated in the S-M/A sensing film, the uniformly distributed strong S–Ag bonds forced the cracks to deflect and twist[19,57,58]. The high density of S–Ag coordination bonds causes large numbers of cracks to deflect and twist, which is quite different from the straight cracks observed in the M/A film. Some deflected cracks in the S-M/A film may intersect with each other during the propagation under strain, thus inducing the formation of a mesh-like crack pattern in the sensing film[23]. Under a 10% applied strain, there were substantially shorter crack lengths (25–35 µm) and much smaller islands (2–3 × 10$^3$ µm$^2$) in the S-M/A film than those in the M/A sensing film (240–300 µm in length, dimensions of 2.5–3.5 × 10$^4$ µm$^2$) (Fig. 5c, g, and k–n). Such short cracks and small islands allowed the S-M/A film to significantly reduce its dimensions while maintaining good detectability to large strain. ABAQUS finite element simulation results also confirmed that the M/A sensing film had a uniform strain distribution, resulting in cut-through cracks, while the S-M/A film exhibited a strain-modulating effect that contributed to the formation of short and mesh-like cracks (Fig. 5j)[5].

Upon increasing the applied strain to 30% (Fig. 5d, h), large gaps (2–6 µm in width) appeared between matching crack edges in the M/A film, while the crack gaps in the S-M/A film remained at the nanometer-scale (<10 nm), which prevented premature failure of the sensing film[5,20]. This successful retention of nanocracks in the S-M/A film, even under large strain, was attributed to the presence of dynamic and reversible S–Ag bonds in the sensing film. The deformation/reformation of S–Ag coordination bonds between the S-MXene and AgNW dissipated energy and buffered crack energy in the S-M/A film upon stretching[29,30]. When the S-M/A film was subjected to longitudinal

strain, adjacent crack edges from neighboring islands converged due to transverse compression, which provided a pathway for electron tunneling[51] (Supplementary Fig. 19). Thus, the high strain sensitivity of S-M/A over a large strain range may have originated from a combination of ohmic conduction and tunneling transport variations[51].

Due to S–Ag bond breakage, upon further stretching the sensing film to 60% strain, the layered MXene nanosheets and AgNW began to slide over each other[25]. The synergistic effect of S–Ag bond breakage and layer slippage allowed the S-M/A film to further uniformly accommodate the applied stress over the entire sensing film, which greatly mitigated the crack gap opening size. As shown in Fig. 4e and Fig. S17, the crack gap opening in the S-M/A sensing film under 60% strain remained quite small (mostly smaller than 1 μm in width), and some edges from neighboring conductive islands remained in contact with each other. In contrast, parallel cut-through cracks with a gap width of about 10 μm were visible on the M/A film (Fig. 5i and Supplementary Fig. 18b). The magnified SEM images in Fig. 4e, i further highlight the different crack morphologies generated in the S-M/A and M/A films under 60% strain (Supplementary Figs. 17 and 18). The crack in the S-M/A film clearly exhibited a high degree of MXene and AgNW sliding, which bridged the gaps of microcracks, whereas the crack edges in the M/A film completely separated from each other.

The synergistic effect of the strong and dynamic S–Ag bonds is further illustrated in the SEM images at 100% strain (Supplementary Fig. 20a) and after 5000 stretch-release cycles to 60% strain (Supplementary Fig. 20b). Most cracks in the S-M/A film were compressed, with a small island area, a small gap opening, and some edges connected even under a large strain and after long-term strain cycles. Thus, our proposed crack modulation strategy by introducing strong, dynamic, and reversible S–Ag coordination bonds plays a critical role in achieving a high sensitivity, wide strain range, high stability, and minimal size in the S-M/A sensing film.

The high sensing performance and small sensing area of the S-M/A sensor allowed us to develop a strain sensor array with high temporospatial resolution. By incorporating scanning electrodes (~40 μm in line width) and interconnector lines through direct ink writing, we constructed two strain sensing arrays with device densities of 100 (10 × 10) S-M/A$_{0.5}$ (Supplementary Fig. 21) and 36 (6 × 6) S-M/A$_1$ sensing units per cm$^2$, respectively (Fig. 6a). To demonstrate the practical applications of these dense and stretchable strain sensor arrays, we first applied the 36-channel sensor array as a wearable multichannel pulse monitoring system to provide complete information about the temporospatial dimensions of a pulse waveform for intelligent health care applications (Fig. 6a and Supplementary Fig. 22). In the past decade, various sensing techniques, particularly resistive-type strain sensors, have been developed as wearable sensor systems for continuously monitoring pulse signals[12,25,59]. However, reported sensors were usually assembled from strain sensor units with large footprints (>10 mm$^2$)[14,15,60]. Such large dimensions inevitably introduce substantial practical problems in terms of pulse monitoring, such as positioning difficulty, pulsing information loss, and poor reliability and accuracy[14]. In clinical practice, it is desirable to develop wearable strain sensor arrays with a high temporospatial resolution, high sensitivity, wide working strain range, and high reliability.

According to Shang Han Lun, a book on traditional Chinese medical science, the radial artery pulse range is about 1 cm in width and 3 cm in length with three locations of Cun, Guan, and Chi (Fig. 6a)[15]. Thus, we attached a 36-channel strain sensor array with dimensions of 1 × 1 cm$^2$ at the Cun position to acquire the temporal and spatial dimensions of a pulse wave (Supplementary Fig. 22). Figure 6b shows the pulse response of channel 3c over 80 s from a 25-year-old male volunteer. Figure 6c displays an enlarged section of the pulse waves in Fig. 6b, in which the early systolic peak (P), point of inflection (T), and dicrotic notch and dicrotic peak (D) are distinguishable[14,15,21,61]. To measure the spatial dimensions of the pulse waveforms, a column

diagram composed of 36 points is displayed in Fig. 6d, where each point represents the pulse intensity measured by the corresponding sensing unit in the sensor array. The x-axis corresponds to the pulse width and sensing unit positions from 1 to 6, the y-axis represents the pulse length and sensing unit positions from a to f, and the z-axis shows the pulse intensity at each point (Fig. 6a). The surface fitting method was utilized for surface smoothing and to connect the signals from each sensing channel to provide a pulse strength mapping surface. To examine the temporal dimensions of the pulse wave, the real-time, dynamic display of the 3D pulse strength distribution measured from 36 channels for every pulse cycle was realized using a customized LabVIEW program. The results are presented in Supplementary Movie 1, and the 3D shapes of the fitted surface reflecting the pulse intensity every 0.1 s over one complete pulse cycle are presented in Fig. 6e. The pulse intensity distribution, pulse length, pulse width, and pulse shape were characterized. The intensity distribution map at characteristic peaks P, T, and D were distinguished at 0.3 s, 0.6 s, and 0.8 s.

Moreover, a small object with a complex shape was placed on the 36-channel strain sensor array to carry out the strain mapping test (Supplementary Fig. 23a). The output signals of the S-M/A$_1$ sensing units in the array could accurately distinguish minute morphology differences and mapped the strain distribution of the object (Supplementary Fig. 23b, 23c). A pipette with a tiny tip was also poked on the sensing array (Supplementary Fig. 23d). The 3D map of intensity distribution corresponding to the tiny tip shape was clearly distinguished (Supplementary Fig. 23e). All these results demonstrate the great potential for applying our high-density sensing arrays in sophisticated applications that require both high sensing accuracy and temporospatial resolution.

## Discussion

In summary, we developed a stretchable, ultrasensitive, and minimal crack-based strain sensor by leveraging a molecular-level crack modulation strategy to introduce strong, dynamic, and reversible S–Ag coordination bonds in an inorganic-matrixed conductive film with alternating layers of S-MXene nanosheets and AgNW networks. The strong S–Ag bonds regulated the local strain field and formed high-density mesh-like cracks that made it possible to minimize the sensing area while also maintaining a high sensitivity. The dynamic S–Ag bonds dissipated energy and prohibited crack gaps from widening, thus guaranteeing a large stretchability and wide strain range for the sensing film. The reversible S–Ag bonds imparted a healing effect to the sensing structure during repeated strain-release cycles, thus improving the sensing stability and reliability. As a result, we overcame the inherent tradeoff between sensing area and sensing performance commonly observed in crack-based strain sensors and achieved a minimal strain sensing element with ultrahigh sensitivity, ultrawide strain range, small hysteresis, fast response speed, and long-term stability. These features, together with the simple and scalable fabrication process, enabled the construction of a high-density stretchable strain sensor array with high temporospatial resolution.

## Methods

### Materials

Ti$_3$AlC$_2$ powder was purchased from Laizhou Kai Kai Ceramic Materials Co., Ltd. AgNW was synthesized according to our previous report, with an average diameter and length of about 20 nm and 20 μm, respectively[62]. Polyurethane was purchased from Yantai Wanhua Polyurethane Co., Ltd. MPTES and the curing agent (1173) was purchased from Alfa Aesar.

### Characterization

The strain sensor was elongated using a motorized linear stage equipped with a built-in controller (Red Star Yang Technology), while

the resistance was measured using a Keithley 2000 digital multimeter. To test the sensing films in the stretched state, samples were initially extended to a specific strain before being adhered to the sample stage using a thermally curable adhesive to ensure the constant deformation of the samples throughout testing. SEM characterization was performed using a JSM-7800 field-emission scanning electron microscope with an accelerating voltage of 5.0 kV. The electric conductivity of samples was measured by a digital and intelligent four-probe meter (ST2258C). XRD patterns were measured by Smartlab-3KW + UltimaIV (3KW). XPS was performed on a Thermo Scientific ESCALAB 250Xi spectrometer. FT-IR spectra were measured using an IR spectrometer (BRUKER VECTOR22) over the wavenumber range of $4000–400\ cm^{-1}$. UV-vis spectra were obtained using a Shimadzu UV-2700 spectrophotometer. All measurements were performed at room temperature (~25 °C). The sensing array pattern was produced via an LPKF Proto-Laser R4 (LPKF Laser & Electronics AG, power = 8 W) with a pulsed laser ($\lambda = 515\ nm$) at a repetition rate of 100 Hz. By modulating the laser spot size and pulse intensity, the etching width and depth were varied. Thermogravimetric analyses were performed using a Labsys EVO Setaram instrument (Setaram Instrumentation, Caluire, France). Approximately 5 mg of each sample were weighed in an open alumina crucible (Setaram Instrumentation, Caluire, France) and heated from 50 °C to 700 °C in a He atmosphere (20 mL/min) with a heating rate equal to 10 °C/min.

## Simulation

This study utilized ABAQUS finite element software to construct a three-dimensional finite element model of S-M/A and M/A sensing films. The model included upper and lower layers of S-MXene nanosheets and a middle layer of AgNW. The mass density, elastic modulus, and tensile strength of AgNW were $10.5\ g\ cm^{-3}$, 86 GPa, and 2 GPa, respectively. The mass density, elastic modulus, and tensile strength of MXene (or S-MXene) were $4.4\ g\ cm^{-3}$, 80 GPa, and 0.67 GPa, respectively. The Ag–S coordination bond energy was $17.74\ kcal\ mol^{-1}$ [28]. Constraints were applied on both sides, and a strain of 10% was applied in the $x$-direction. A tetrahedral free mesh was adopted for grid partitioning, and the model had a total of 76,916 units.

## Preparation of MXene ($Ti_3C_2T_x$) nanosheets

$Ti_3C_2T_x$ samples were synthesized by selectively etching aluminum from titanium aluminum carbide ($Ti_3AlC_2$, 400 mesh size). Initially, 1 g of LiF was dissolved in 20 mL of 9 M HCl and stirred for 5 min. Subsequently, 1 g of $Ti_3AlC_2$ was gradually added to the solution, and the reaction was allowed to proceed under constant stirring at 35 °C for 24 h. After etching, the resulting suspension was subjected to centrifugation at $1370\times g$ for 5 min and then rinsed with deionized water, which was repeated until the supernatant pH exceeded 6. The mixture was then mechanically shaken for 30 min and centrifuged for an additional 30 min to separate the supernatant. MXene powder was obtained after freeze-drying the collected supernatant.

## Preparation of S-MXene

To prepare the S-MXene, MPTES was initially introduced into a water-ethanol (the volume ratio of water to ethanol is 5:1) mixture and hydrolyzed under magnetic stirring for 30 min. Upon complete dissolution, the resulting solution was combined with the MXene dispersion and heated for 2 h at 60 °C. The MXene:MPTES mass ratio was maintained at 100:20. Following the reaction, the suspension was centrifuged at $11180\times g$ for 10 min. The supernatant was removed and $N,N$-dimethylformamide (DMF) was added to re-disperse the remaining precipitate. These centrifuging and washing steps were repeated three times to wash off the unreacted MPTES and oligomer. After that, the remaining precipitate was dispersed in DMF again and centrifuged at $112\times g$ for 3 min. Finally, the supernatant that contained S-MXene was collected, and the remaining precipitate that contained a by-product of poly(MPTES) was discarded (Fig. S24). The content of MPTES grafted onto the S-MXene was about 3.6 wt% (Fig. S25).

## Fabrication of S-M/A sensor elements

To fabricate the S-M/A strain sensor, a glass slide was thoroughly cleaned using detergent and deionized water. Then, S-MXene and AgNW dispersions were prepared with concentrations of 2.5 mg/mL and 1 mg/mL, respectively, which were then alternately deposited via spin coating to achieve layer-by-layer assembly. Specifically, the S-MXene dispersion was dropped onto the glass slide, spin-coated at $112\times g$ for 15 s, and dried. Subsequently, 100 μL AgNW dispersion was dropped onto the S-MXene film, spin-coated at $112\times g$ for 15 s, and dried. After repeating these steps three times, the films were thermally treated on a hot plate at 70 °C for 10 min. The existence of strong S–Ag coordination bonds between AgNW and S-MXene is vital to forming uniform and multilayered S-M/A films through LBL technology [63]. Finally, a mixture of PU with 1% curing agent (1173) was applied and cured, and then the S-M/A strain sensor was peeled off from the glass slide and cut into test samples. Thus, the S-M/A strain sensor was partially embedded into the PU substrate. With rich carbamate functional groups, PU substrate could form hydrogen bonds with S-M/A sensing material. The partially embedded structure together with the abundant hydrogen bonds ensured good adhesion between PU substrate and S-M/A sensing film. Control groups were also prepared following the same procedure as above. The initial resistance for S-M/A sensors was ~15 Ohm.

## Fabrication of S-M/A sensor array

To construct a sensor array, conductive carbon nanotubes were printed on the edges of the $S-M/A_{0.5}$ and $S-M/A_1$ sensing elements and the center of the interspaces to serve as electrodes. This was accomplished via direct ink writing with a line width of 40 μm. To prevent short-circuiting, a thin layer of polyurethane was printed and cured as an insulating layer at the junction of the two electrodes before the second electrode layer was printed. The device fabrication was completed by pouring polyurethane over the device and curing it.

## Ethical statement

The participants in the study are fully informed about the purpose, risks, and benefits of the research. This study was approved by the Administration Committee of Tianjin University of Science and Technology, Tianjin Province, China.

## Reporting summary

Further information on research design is available in the Nature Portfolio Reporting Summary linked to this article.

# Data availability

The data that are necessary to interpret, verify, and extend the research in the article are provided in the main text and/or SI. The experiment data that support the findings of this study are available from the corresponding authors upon request. Source data are provided with this paper.

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

## Acknowledgements

This work was supported by National Natural Science Foundation of China (52173238), the Municipal Natural Science Fund of Tianjin (20JCJQJC00010 and 21JCQNJC00170), the Ministry of Science and Technology of China (2022YFA1203304), and the Fundamental Research Funds for the Central Universities of Nankai University (63191520), and the Tianjin Graduate Research and Innovation Project (2022SKYZ069, 2022SKYZ141).

## Author contributions

Y.L., Z.X. and X.J. contributed equally to this work. J.L. and B.C. supervised the project. J.L. and Y.L. conceived and designed the research. Y.L., Z.X., X.J., X.X., F.C., X.P., Z.F, Y.C., Z.Z. and H.L. participated in materials preparation, device fabrication, device test, or interpretation of results. J.L. and Y.L. analyzed the data and co-wrote the manuscript. All authors analyzed and discussed the results.

## Competing interests

The authors declare no conflict of interest.
