## [Peer Review File · Nature Communications]

Ag-Thiolate Interactions to Enable an Ultrasensitive, Stretchable, and Minimal Strain Sensor with High Temporospatial ResolutionREVIEWER COMMENTS

Reviewer #1 (Remarks to the Author):

In this work, a sensing film was fabricated from thiolate-terminated MXene (S-MXene) nanosheets and silver nanowire (AgNW) via a layer-by-layer technique to form dynamic thiol-silver (S-Ag) coordination bonds that were distributed in the multilayered sensing thin-film, and it shows good performance. This work is undoubtedly of some value for those who are working in the field of strain sensors. However, the following issues should be addressed:

1. During the preparation process of S-MXene, does MPTES self-polymerize? How to remove self-polymerization by-products? How much MPTES is actually grafted onto MXene? The FT-IR in Fig.2b shows basically the characteristic peaks of MPTES. The structure of S-MXene needs further characterization and discussion.
2. With strong thiol-silver (S-Ag) coordination bonds, how to achieve uniform dispersion of AgNW on the S-MXene coating through spin coating when preparing S-M/A sensing film by layer-by-layer technology? The strong S-Ag coordination bonds interaction is not conducive to dispersing AgNW and S-MXene. The authors need to conduct further characterization (such as TEM and EDS mapping) and strengthen the discussion.
3. MXene is prone to oxidation in a water-oxygen environment, leading to performance degradation. Did the author consider this issue during the preparation process?
4. As a sensor substrate, the modulus of polyurethane is very important. If it doesn't match the S-M/A, it may cause interface detachment and affect the sensing performance. In addition, the author should discuss the adhesion between S-M/A and PU.
5. In Figure 4 SEM images, the distribution of AgNW and S-MXene cannot be clearly observed. The author should add element mapping tests. Also, the comparative discussion of M/A sensing film and S-M/A sensing film should be strengthened.
6. The thickness of S-M/A sensing film is a key factor affecting sensing performance. The article only discusses the relationship between area and sensing performance, but lacks a discussion of the impact of thickness. The characterization of S-M/A sensing film thickness in the supplementary material should use a unified scale. The interaction between AgNW and S-MXene layers also needs to be characterized by element mapping.
7. What is the fatigue stability of S-M/A sensors? Does the air humidity have an impact on sensing performance?

Reviewer #2 (Remarks to the Author):

The authors have designed a strain sensor with a small area but excellent performance through mesh-like cracks by means of Ag-S coordination bonds. This sensor has outstanding performance in several metrics. This is a pretty good piece of work, but there are still a couple of issues that need to be revised.

1. The author claims " The sensitivity (gauge factor (GF)) in the small-strain range of 0–0.05% for S-M/A0.5 reached about 500 with a linearity of 0.99 (Figure 3c), which is similar to that of S-M/A1, S-M/A2, and S-M/A5 sensors with larger sensing areas (Figure S4)". But there is a GF of 700, and it is inappropriate to say that they are similar.
2. The author claims " To confirm repeatability, more than 10 samples for each device were tested and exhibited similar performance.". It is recommended that the authors give specific data
3. What is the initial resistance of the sensor
4. Why is the density of data points taken during the testing of sensing performance different? For example, Fig.S4b Fig.S6a
5. Fig.S7 should be looped several times to characterise the hysteresis of the sensor
6. The sensors are prepared by layer-by-layer method, does the number of layers have an effect

on the performance and the form of crack extension

7. Please clarify in more detail why S-M/A produces mesh-like cracks, which the authors have not articulated clearly here.

8. There are many sensors that achieve the detection of pulse, to better highlight the excellent performance of this sensor, it is suggested to add brighter demonstration experiments.

9. In the simulation section, the authors used the modulus of MXene, should they have used the modulus of the MXene film? Also the description of the simulation process is not detailed enough.

RESPONSE TO REVIEWERS' COMMENTS

A list of changes, including:

1. We have added more experimental details about how to remove self-polymerization by-products in the method part of the revised manuscript and SI (highlight on Page 22)
2. We have added TGA and DTG measurements to determine how much MPTES was grafted onto MXene (highlight on Page 22 and Figure S24 and S25)
3. We added FT-IR spectrum for MPTES in Fig. 2b and XPS characterization for S-MXene in the revised manuscript and SI (highlight on Page 8, Figure 2b and Figure S2)
4. We have added SEM, TEM and EDS mapping characterizations and discussions about the uniform distribution of AgNW and S-MXene in the revised manuscript and SI (highlight on Page 8, Figure S6 and S7).
5. We have added the experiments to monitor the resistance changes of MXene and S-MXene films at 80 °C and RH of 85%, and the corresponding discussion in the revised manuscript and SI (highlight on Page 8 and Figure S3).
6. We have added the corresponding discussions about the adhesion between S-M/A and PU in the experimental of the revised manuscript (highlight on Page 22)
7. We have added magnified SEM images with corresponding element mapping for the S-M/A and M/A samples under 0% and 60% strain in the revised SI (highlight on Figure S7, S17 and S18). The corresponding discussion has been added in the revised manuscript (highlight on Page 14).
8. We have modified the SEM characterization of S-M/A sensing film thickness using a unified scale in the revised SI (Figure S4).
9. We have added the measurement and the corresponding discussion of the sensing performance of S-M/A₁ sensing films with different thickness (0.7 μm, 1.4 μm, and 2.4 μm) in the revised manuscript and SI (highlight on Page 8, Supplementary Note 1 and Figure S5).
10. We have added the element mapping characterizations to confirm the uniform distribution of AgNW on S-MXene in the revised SI (highlight in Figure S7, S17 and S18).
11. We have added the fatigue stability of S-M/A₅ under 70% strain for 5000 cycles under RH of 85% in the revised SI (highlight in Figure S16) and added corresponding discussion in the revised manuscript (highlight on Page 12).

12. We have added the corresponding discussion about the adhesion between S-M/A and PU (highlight on Page 23).
13. We have modified the Figure S9a in the revised SI.
14. We have modified the statement “5 samples for each device were tested and exhibited similar performance” in the revised manuscript (highlight on Page 11) and added the data for the other four sample devices, including sensing performance and response time for S-M/A_{0.5} sensors, in the revised SI (highlight in Figure S10 and S11).
15. We have added the initial sheet resistance of the S-M/A sensor in the revised manuscript (highlight on Page 23).
16. We have replaced old data with new ones that had similar density of data points (highlight in Figure S9 and S13).
17. We have added the relative resistance variation as a function of applied strain for S-M/A₅, S-M/A₂, S-M/A₁, and S-M/A_{0.5} sensors under the 1st, 10th, and 100th stretch-release cycle in the revised manuscript and SI (highlight on Page 11 and Figure S14).
18. We have added more discussion about the generation of mesh-like cracks in the revised manuscript (highlight on Page 14).
19. We have added the application about the sensing array to detect an object with complex shape and a tiny pipette tip in the revised manuscript and SI (highlight on Page 19 and Figure S23).
20. We have added the detailed description of the simulation process in the revised SI (highlight in Supplementary Note 2).

Reply to the reviewer#1:

Comments 1: In this work, a sensing film was fabricated from thiolate-terminated MXene (S-MXene) nanosheets and silver nanowire (AgNW) via a layer-by-layer technique to form dynamic thiol-silver (S-Ag) coordination bonds that were distributed in the multilayered sensing thin-film, and it shows good performance. This work is undoubtedly of some value for those who are working in the field of strain sensors.

REPLIES: Many thanks for the positive comments!

Comments: During the preparation process of S-MXene, does MPTES self-polymerize? How to remove self-polymerization by-products? How much MPTES is actually grafted onto MXene? The FT-IR in Fig.2b shows basically the characteristic peaks of MPTES. The structure of S-MXene needs further characterization and discussion.

REPLIES: Thanks for the question and suggestions! The MPTES would self-polymerize during the fabrication process of S-MXene (*Nat. Commun.* 2022, 13, 1119). We have added more experimental details about how to remove self-polymerization by-products in the method part of the revised manuscript and SI (highlight on Page 22). Moreover, we carried out thermogravimetric analysis (TGA) and thermogravimetry (DTG) measurements to determine how much MPTES was grafted onto MXene. As shown in Figure S24 and S25 in the revised SI, the content of MPTES grafted onto the S-MXene was about 3.6 wt% (highlight on Page 22 and in Figure S24 and S25). As shown in FT-IR spectrum of Fig. 2b, the new peak appeared at 944 cm^{-1} was ascribed to Ti-O-Si asymmetric stretching from the S-MXene (*Chem. Eng. J.* 2020, 383, 123125; *Nat. Commun.* 2022, 13, 1119). No such peak could be seen in MPTES and pure MXene (highlight in Figure 2b). This confirmed the successful grafting of MPTMS onto MXene. Moreover, we added XPS characterization for S-MXene in the revised manuscript and SI (highlight on Page 8 and Figure S2). It revealed a typical Ti-O-Si bond at 531.7 eV in O 1s spectra (highlight in Figure S2a) and Si-O-C at 102.2 eV in Si 2p spectra (highlight in Figure S2b), which also confirmed the successful grafting of MPTES onto MXene (*Nat. Commun.* 2022, 13, 1119.; *Physics Procedia*, 2012, 32, 95).

Comments: With strong thiol-silver (S-Ag) coordination bonds, how to achieve uniform dispersion of AgNW on the S-MXene coating through spin coating when preparing S-M/A sensing film by layer-by-layer technology? The strong S-Ag coordination bonds interaction is not conducive to dispersing AgNW and S-MXene. The authors need to conduct further characterization (such as TEM and EDS mapping) and strengthen the discussion.

REPLIES: Thanks for the question! In fact, in the layer-by-layer technology, the assembly materials need to have strong affinity and intermolecular interactions, such as electrostatic interaction (*Adv. Mater.* 2001, 13, 11), supramolecular interactions (*J. Am. Chem. Soc.* 2005, 127, 20, 7594), and coordination bond (*Chem. Mater.* 2003, 15, 196; *Thin Solid Films.* 2016, 600: 76; *Chem. Commun.* 2007, 14, 1395), to ensure the successfully initiate film growth and construction (*Chem. Rev.* 2014, 114, 8883). Thus, the existence of strong S-Ag coordination bonds between AgNW and S-MXene is vital to form the uniform and multilayered S-M/A films through LBL technology (*Chem. Rev.* 2014, 114, 8883). In our LBL process, the S-MXene dispersion was spin-coated on the completely dried AgNW film and vice versa. We did not mix these two dispersions together, thus would not affect their dispersity. We have added SEM, TEM, and EDS mapping characterizations and discussions about the uniform distribution of AgNW and S-MXene in the revised manuscript and SI (highlight on Page 8, 22 and in Figure S6 and S7).

Comments: MXene is prone to oxidation in a water-oxygen environment, leading to performance degradation. Did the author consider this issue during the preparation process?

REPLIES: Thanks for the questions! The MPTES was hydrolyzed in the presence of water to create the corresponding hydroxysilane, which can react with the -OH surface groups of MXene to form a covalent bond (Ti-O-Si) through a silylation reaction (highlight on Figure S24). Thus, surface functionalization in the S-MXene could block the contact between MXene and water or dissolved oxygen and prohibited the oxidation reaction (*ACS Nano* 2020, 14, 11490; *Adv. Mater.* 2022, 34, 2107554; *FlatChem.* 2019, 17, 100128; *J. Energy Chem.* 2023, 87, 439; *Anal. Chem.*

2022, 94, 20, 7319; *EcoMat.* 2023, 5, 10, e12395). We added the experiments to monitor the resistance changes of MXene and S-MXene films at 80 °C and relative humidity (RH) of 85% over 30 days. Compared with the pure MXene film, the S-MXene film exhibited much improved oxidation resistance. We have added the corresponding discussion and data in the revised manuscript and SI (highlight on Page 8 and Figure S3 and S24).

Comments: As a sensor substrate, the modulus of polyurethane is very important. If it doesn't match the S-M/A, it may cause interface detachment and affect the sensing performance. In addition, the author should discuss the adhesion between S-M/A and PU.

REPLIES: Thanks for the suggestion! In the final step of fabrication process for S-M/A sensor elements, a mixture of PU with 1% curing agent was applied and cured on the S-M/A sensing material, and then the S-M/A strain sensor was peeled off from the glass slide and cut into test samples. Thus, the S-M/A strain sensor was partially embedded into the PU substrate. With rich carbamate functional groups, PU substrate could form hydrogen bonds with S-M/A sensing material. The partially embedded structure together with the abundant hydrogen bonds ensured good adhesion between PU substrate and S-M/A sensing film. We have added the corresponding discussions in the experimental of the revised manuscript (highlight on Page 22).

Comments: In Figure 4 SEM images, the distribution of AgNW and S-MXene cannot be clearly observed. The author should add element mapping tests. Also, the comparative discussion of M/A sensing film and S-M/A sensing film should be strengthened.

REPLIES: Thanks for the suggestions! Magnified SEM images with corresponding element mapping for the S-M/A and M/A samples under 0% and 60% strain have been added in the revised SI (highlight in Figure S7, S17 and S18). Uniform distribution of AgNW network can be clearly seen in both S-M/A and M/A samples. The crack gap opening in the S-M/A sensing film under 60% strain remained quite small (mostly smaller than 1 μm in width), and some edges from neighboring conductive islands remained in contact with each other. In contrast, parallel cut-

through cracks with a gap width of about 10 μm were visible on the M/A film. The strong S-Ag coordination bonds can induce stress concentration in the S-M/A sensing film under strain (*Adv. Mater.* 2022, 35, 2208568). The cracks propagate straight through the weak interfaces of M/A sensing film, while in the S-M/A sensing film, the stress-concentrated sites can force the cracks deflect and twist. The high density of uniformly distributed S-Ag coordination bonds causes large numbers of cracks to deflect and twist. Some deflected cracks may intersect with each other during the propagation, thus inducing a mesh-like crack pattern (*Adv. Mater.* 2022, 35, 2208568). We have added the corresponding discussion in the revised manuscript (highlight on Page 14).

Comments: The thickness of S-M/A sensing film is a key factor affecting sensing performance. The article only discusses the relationship between area and sensing performance, but lacks a discussion of the impact of thickness. The characterization of S-M/A sensing film thickness in the supplementary material should use a unified scale. The interaction between AgNW and S-MXene layers also needs to be characterized by element mapping.

REPLIES: Many thanks for the suggestions! We have modified the SEM characterization of S-M/A sensing film thickness using a unified scale in the revised SI (highlight in Figure S4). We also added the measurement of the sensing performance of S-M/A₁ sensing films with different thickness (0.7 μm , 1.4 μm , and 2.4 μm) in the revised manuscript and SI (highlight on Page 8, Supplementary Note 1 and Figure S5). Figure S5 plots the relative change in resistance of the S-M/A₁ strain sensor with various thicknesses (T) of 2.4, 1.4, and 0.7 μm as a function of applied strain. Obviously, the stretchability of the S-M/A₁ strain sensor increases with the device thickness decreases from thickness of 2.4 to 1.4 μm . The highest GF for the rigid S-M/A₁ strain sensor with device thickness of 2.4 μm was calculated to be about 445000 in working strain range of 9-11%. Although the maximum GF drops to about 71400 as the device thickness declines to 1.4 from 2.4 μm , the working strain range shows a significant increase from 11% to 45%. This stretchability increase can be attributed to the fact that reducing thickness in the film can render the brittle thin film flexible (*Sci. Rep.* 2017, 7, 4011). With the thickness further decreased to 0.7 μm from 1.4 μm , the stretchability of the S-M/A₁ strain sensor decreased and the GF values increased. This phenomenon was mainly attributed to easier destruction of electrical junctions between adjacent

sensing islands under stretching when the sensing film had a very small thickness (*J. Mater. Chem. A*, 2020, 8, 10310). We have added the corresponding discussion and data in the revised manuscript (highlight on Page 8) and SI (highlight in Supplementary Note 1 and Figure S5).

As discussed above, the element mapping characterizations have also been added in the revised SI (highlight in Figure S7, S17, and S18).

Comments: What is the fatigue stability of S-M/A sensors? Does the air humidity have an impact on sensing performance?

REPLIES: Thanks for the suggestions! We have demonstrated the fatigue stability in Figure 3h (S-M/A_{0.5}, 1000 cycle@15%), Figure S15 (S-M/A_{0.5}, 4000 cycle@0.05%), and Figure S16a (S-M/A₅, 8000 cycle@70%) in the original manuscript and SI. The relative humidity (RH) for these stability tests was 50%. We also added the fatigue stability of S-M/A₅ under 70% strain for 5000 cycles under RH of 85% in the revised SI (highlight on Figure S16b) and added corresponding statements in the revised manuscript (highlight on Page 12). The S-M/A sensors exhibited a high sensing stability even under RH of 85%.

Reply to the reviewer#2:

Comments: The authors have designed a strain sensor with a small area but excellent performance through mesh-like cracks by means of Ag-S coordination bonds. This sensor has outstanding performance in several metrics. This is a pretty good piece of work, but there are still a couple of issues that need to be revised.

REPLIES: Thanks for the positive comments!

Comments: The author claims " The sensitivity (gauge factor (GF)) in the small-strain range of 0–0.05% for S-M/A0.5 reached about 500 with a linearity of 0.99 (Figure 3c), which is similar to that of S-M/A1, S-M/A2, and S-M/A5 sensors with larger sensing areas (Figure S4)". But there is a GF of 700, and it is inappropriate to say that they are similar.

REPLIES: Many thanks for pointing out this problem! We have modified this in the revised SI (highlight in Figure S9a).

Comments: The author claims " To confirm repeatability, more than 10 samples for each device were tested and exhibited similar performance." . It is recommended that the authors give specific data.

REPLIES: Thanks for the suggestion! We have modified the statement “5 samples for each device were tested and exhibited similar performance” in the revised manuscript (highlight on Page 11). The data for the other four sample devices, including sensing performance and response time for S-M/A_{0.5} sensors, were added in the revised SI (highlight in Figure S10 and S11).

Comments: What is the initial resistance of the sensor.

REPLIES: Thanks for the question! The initial sheet resistance of S-M/A was ~15 Ohm. We have added the resistance of the S-M/A sensor in the revised manuscript (highlight on Page 23).

Comments: Why is the density of data points taken during the testing of sensing performance different? For example, Fig.S4b Fig.S6a.

REPLIES: Thanks for pointing out this problem! We have replaced old data with new ones that had similar density of data points (highlight on Figure S9b and S13a).

Comments: Fig.S7 should be looped several times to characterise the hysteresis of the sensor.

REPLIES: Thanks for the suggestion! We have added the relative resistance variation as a function of applied strain for S-M/A₅, S-M/A₂, S-M/A₁, and S-M/A_{0.5} sensors under the 1st, 10th, and 100th stretch-release cycle (highlight on Page 11 and in Figure S14). All the data of S-M/A sensors showed good repeatability and relatively low hysteresis.

Comments: The sensors are prepared by layer-by-layer method, does the number of layers have an effect on the performance and the form of crack extension.

REPLIES: Thanks for the question! We added the measurement of the sensing performance of S-M/A₁ sensing films with different thickness (0.7 μm, 1.4 μm, and 2.4 μm) in the revised manuscript and SI (highlight on Page 8, Supplementary Note 1, and Figure S5). The thickness of the sensing film increased as the number of coating layer increased. Figure S5 plots the relative change in resistance of the S-M/A₁ strain sensor with various thicknesses (T) of 2.4, 1.4, and 0.7 μm as a function of applied strain. Obviously, the stretchability of the S-M/A₁ strain sensor increases with the device thickness decreases from thickness of 2.4 to 1.4 μm. The highest GF for the rigid S-M/A₁ strain sensor with device thickness of 2.4 μm was calculated to be 445000 in working strain range of 9-11%. Although the maximum GF drops to 71400 as the device thickness declines to 1.4

from 2.4 μm , the working strain range shows a significant increase from 11% to 45%. This stretchability increase can be attributed to the fact that reducing thickness in the film can render the brittle thin film flexible (*Sci. Rep.* 2017, 7, 40116). With the thickness further decreased to 0.7 μm from 1.4 μm , the stretchability of the S-M/A₁ strain sensor decreased and the GF values increased. This phenomenon was mainly attributed to easier destruction of electrical junctions between adjacent sensing islands under stretching when the sensing film had a very small thickness (*J. Mater. Chem. A*, 2020, 8, 10310). We have added the corresponding discussion and data in the revised manuscript (highlight on Page 8) and SI (highlight in Supplementary Note 1 and Figure S5).

Comments: Please clarify in more detail why S-M/A produces mesh-like cracks, which the authors have not articulated clearly here.

REPLIES: Thanks for the suggestion! The strong S-Ag coordination bonds can induce stress concentration in the S-M/A sensing film under strain (*Adv. Mater.* 2022, 35, 2208568). The cracks propagate straight through the weak interfaces of M/A sensing film, while in the S-M/A sensing film, the stress-concentrated sites can force the cracks deflect and twist. The high density of uniformly distributed S-Ag coordination bonds causes large numbers of cracks to deflect and twist. Some deflected cracks may intersect with each other during the propagation, thus inducing a mesh-like crack pattern (*Adv. Mater.* 2022, 35, 2208568). We have added more discussion about the generation of mesh-like cracks in the revised manuscript (highlight on Page 14).

Comments: There are many sensors that achieve the detection of pulse, to better highlight the excellent performance of this sensor, it is suggested to add brighter demonstration experiments.

REPLIES: Thanks for the suggestion! We agree that many strain sensors have been reported to achieve the detection of pulse. However, the reported sensors were usually assembled from strain sensor units with large footprints ($>10 \text{ mm}^2$). Such large dimensions inevitably introduce substantial practical problems in terms of pulse monitoring, such as pulsing information loss, and

poor reliability and accuracy (*Sci. Adv.* 2023, 9, eadh0615). In clinical practice, it is desirable to develop wearable strain sensor arrays with a high temporospatial resolution to achieve the detection of pulse intensity distribution, pulse length, pulse width, and pulse shape. Thus, we applied the 36-channel sensor array as a wearable multichannel pulse monitoring system to provide complete information about the temporospatial dimensions of a pulse waveform for intelligent health care applications. We think this is a good demonstration to highlight the excellent sensing performance of our S-M/A sensing array.

We also added other applications to demonstrate the sensing performance of our S-M/A sensing array in the revised manuscript and SI. Two objects with different shape were placed on the sensing array to carry out the strain mapping test (Figure S23). The output signals of the sensing pixels accurately distinguished minute shape differences and mapped the strain distribution of these two objects. We have added these new data and discussion in the revised manuscript and SI (highlight on Page 19 and Figure S23).

Comments: In the simulation section, the authors used the modulus of MXene, should they have used the modulus of the MXene film? Also the description of the simulation process is not detailed enough.

REPLIES: Thanks for pointing out this problem! In the simulation section, we used the modulus of MXene rather than MXene film since the modulus of MXene film was an uncertain value that could be affected by many factors. We have added more details for the description of the simulation process in the revised manuscript and SI (highlight in Supplementary Note 2).

We conducted tensile and crack distribution analysis on the model using a standard module of ABAQUS software. The simulation process included the following four steps: (1) Material definition. We utilized the Johnson-Cook damage model and referenced stress-strain curves from previous reports for MXene¹ and AgNW². The fracture condition was set based on the tensile properties of MXene and AgNW. The density, elastic modulus, and tensile strength used in the simulation process for MXene (and S-MXene) was 4.4 g/cm³, 80 GPa, and 0.67 GPa, and 10.5 g/cm³, 86 GPa, and 2 GPa, respectively. (2) Contact settings. We then specified frictionless and adhesive contact between AgNW and MXene (or S-MXene). The calculation of Ag-S bonding

points was based on the MPTES molecule weight and S-MXene mass, with the adhesion strength defined as $17.74 \text{ kcal}\cdot\text{mol}^{-1}$. (3) Mesh partitioning. Tetrahedral free mesh partitioning with C3D10 elements for MXene (or S-MXene) and AgNW models was employed. (4) Boundary conditions. We fixed the constraints at one end and the load tensile strain of 10% at the other end.

REVIEWERS' COMMENTS

Reviewer #1 (Remarks to the Author):

The authors have revised the manuscript according to the comments. Therefore, I think it can be accepted and published in Nature Communications.

Reviewer #2 (Remarks to the Author):

The author has made the necessary modifications as requested. The following articles are suggested to be added to the introduction section to make it more comprehensive.

<https://doi.org/10.1038/s41528-024-00301-7>

10.1039/d1mh00384d